# Crash Classification by Congestion Type for Highways

**Tai-Jin Song** [1] , **Sangkey Kim** [2,*] , **Billy M. Williams** [3] , **Nagui M. Rouphail** [4] **and George F. List** [4]

1   Department of Urban Engineering, Chungbuk National University, Chungju 28644, Korea; tj@cbnu.ac.kr
2   Local Investment Management Center, Korea Research Institute of Location Administration, Seoul 06654, Korea
3   Institute for Transportation Research and Education, North Carolina State University, Raleigh, NC 27695, USA; billy_williams@ncsu.edu
4   Department of Civil, Construction, and Environmental Engineering, North Carolina State University, Raleigh, NC 27695, USA; rouphail@ncsu.edu (N.M.R.); gflist@ncsu.edu (G.F.L.)
*   Correspondence: ksangke@krila.re.kr

**Abstract:** Effective management of highway networks requires a thorough understanding of the conditions under which vehicular crashes occur. Such an understanding can and should inform related operational and resource allocation decisions. This paper presents an easily implementable methodology that can classify all reported crashes in terms of the operational conditions under which each crash occurred. The classification methodology uses link-based speed data. Unlike previous secondary collision identification schemes, it neither requires an a priori identification of the precipitating incident nor definition of the precipitating incident's impact area. To accomplish this objective, the methodology makes use of a novel scheme for distinguishing between recurrent and non-recurrent congestion. A 500-crash case study was performed using a 274 km section of the I-40 in North Carolina. Twelve percent of the case study crashes were classified as occurring in non-recurrent congestion. Thirty-seven percent of the crashes in non-recurrent congestion classified were identified within unreported primary incidents or crashes influence area. The remainder was classified as primary crashes occurring in either uncongested conditions (84%) or recurrent congestion (4%). The methodology can be implemented in any advanced traffic management system for which crash time and link location are available along with corresponding archived link speed data are available.

**Keywords:** collision classification; recurrent and non-recurrent congestion; link speed data; traffic safety; advanced traffic management system

---

## 1. Introduction

The primary role of an Advanced Traffic Management Systems (ATMS) is to improve reliability and safety through active real-time traffic management and control. Vehicular crashes endanger lives, damage property, and cause congestion, presenting an obstacle to the goal of improving the safety, efficiency, and sustainability of the transportation system. To manage a system well, it is important to understand the conditions under which crashes happen. This knowledge can inform crash management and resource allocation for incident response. For instance, Variable Speed Limits (VSL) may be an effective countermeasure to prevent crashes during recurring congestion. Conversely, incident response time may be the most effective strategy for mitigating the impacts of crashes during congested periods. However, crash records in a crash analysis database do not indicate congestion conditions at the crash scene but only environmental, crash, roadway, and driver characteristics [1–3].

Another motivation for understanding and classifying crashes is that such information helps improve reliability and safety [4–10]. Many previous studies have focused on identifying the relationship between crashes and the traffic flow rate. The relationship between reliability and safety is less well understood, except that vehicular collisions and other unplanned incidents increase travel time variability and decrease reliability. Operating jurisdictions tend to note simply whether the recurring or non-recurring congestion was extant at the time of the crash [11,12]. Conversely, many traffic safety researchers have studied secondary collision and the factors affecting their occurrence on freeways [13,14].

This analysis focuses on classifying crashes, especially whether they occurred during recurrent and non-recurrent congestion. The implicit objective is to help operating agencies understand how to reduce the number of secondary collisions and mitigate their risk. A necessary precursor is a method to classify each crash in terms of whether or not it occurred during congested conditions, and if so, whether the congestion was the cause of or impacted by the event. This calls for developing an integrated methodology to classify crashes by congestion type.

With these considerations in mind, this paper presents an easily implementable methodology that can classify all reported crashes in terms of the operational conditions under which each crash occurred. It classifies crashes three cases: (1) crash not during congested conditions, (2) crash during non-recurrent congestion, and (3) crash during recurring congestion. Unlike previous secondary collision identification schemes, it requires neither identification of the precipitating incident nor a definition of the precipitating incident's impact area. It supports decision-makers in their efforts to implement both safety and mobility treatments that are precisely targeted and effective.

In what follows, relevant studies are reviewed and knowledge gaps are highlighted. Then, the proposed methodology is described and applied to a 274 km section of I-40 in North Carolina of the United States. The paper concludes with a presentation of the findings, conclusions, and recommendations for further research.

## 2. Literature Review

Several studies have tested the relationship between crash rates and flow rates or density [4–10]. However, the results have been inconclusive. Zhou and Sisiopiku [4] found that there was a U-shaped relationship between V/C ratio and crash rate on freeways. On the other hand, Lord et al. [8] did not find relationships between crash rate and congestion or severity and congestion.

Insofar as crash types and congestion are concerned, there are two important questions: (1) does congestion influence the crash type and (2) vice versa. Most studies have focused on the former but this one was concerned with the latter. Golob et al. [9] and Lee et al. [10] found that rear-end crashes were more likely under unstable traffic flow conditions. Elvik et al. [15] identified main factors influencing accidents on road bridge and found that traffic volume was the most influencing factor. Wang et al. [6] found that traffic congestion had little or no impact on crash rates. However, their statistical model failed to pass statistical significance tests. The reason for this might be that they used a congestion index to test the relationship, and that index was based on the average congestion level across an entire year. More recent studies have focused on identifying the relationship between road accidents and traffic volume. Xu et al. [16] found that high traffic volume was responsible for 25.6% of the serious casualty crashes indicating that there is a positive relationship between traffic volume and road accidents. Zhan-Moodie [17] concluded that congestion can be linked with crashes by superimposing crash areas on top of congested areas using GIS shapefiles. Retallack and Ostendorf [18] reviewed this work and concluded that the method was not tested using congestion information that pertained at the time of the crash. The study presented here addresses this issue by using a congestion measure that is predicated on the traffic conditions at the time of the crash.

At extant traffic condition (congestion) can be classified as either recurrent or non-recurrent. A definition of non-recurrent congestion is as follows; delay caused by an incident, a work zone, adverse weather, or other non-repetitive event [12,19–25]. Chung [21] defines non-recurrent congestion

as the extra delay caused by incidents compared with the annual average section travel speed. For instance, if the free-flow speed is 60 mi/h and annual average section travel speed is 30 mi/h during peak periods, then it is assumed that recurrent congestion occurs.

Defining recurring congestion is more problematic [19–22,26]. Oxford's Dictionary defines "recurrent" as being something that occurs often or repeatedly. In other words, recurrent congestion should be "predictable" in both location and time. Drivers should indicate that "this area, at this time, is often". Most previous studies use either the mean or the median of a speed distribution during a specified time of day to measure recurrent delay and extra delay caused by incidents as non-recurrent congestion on a freeway. Caltrans [27] defines recurring congestion as the combination of a location, time, and speed: for example, an average speed of 35 mile per hour or less for 15 min or more on a specific freeway segment. Schaefer [25] defines it as a Travel Time Index (TTI) of 1.5 or more for a given segment. Song et al. [11] recently employed a data-driven approach using link speeds. Their approach defined "recurrent" consistent with the Oxford's Dictionary definition and identified recurrent bottleneck locations and their time span.

Collision classification tends to rely on deterministic queuing theory. If a specific freeway segment is under study, the rules are often as follows. First, crashes which occur during times of non-recurrent congestion are classified as such. These become the "primary" incidents. Then, the boundaries of the impact area are determined. The rules for doing this may be either static or dynamic. In the static case, one might assume that a secondary incident has arisen if it occurs within 15 min of the primary event and within 1 mile upstream [28]. This rule might be applied even though the primary incident impact area is longer. That is, incidents occurring more than a mile upstream are not classified as secondary.

Dynamic thresholds overcome some of these limitations [23,24,29,30]. Chou and Miler-Hooks [29] created a "simulation-based secondary incident filtering" method (SPSIF). They implemented a regression model to identify the boundaries of the primary incident impact area. All subsequent incidents within that area were classified as secondary. Zhang and Khattak [31] used queuing analysis to study single-pair events (one primary and one secondary incident) and large-scale events (one primary and many secondary incidents). Their objective was to determine the "back of queue" location. Yang et al. [32] proposed the use of historical virtual sensor measurements to identify secondary crashes. They used a Representative Speed Contour Map (RSCM) with percentile speed of historical incident-free virtual sensor speed measurements of each spatiotemporal cell. Moreover, Goodall [13] used private-sector speed data provided from INRIX TMC data for the first analysis of secondary crash occurrence to integrate incident timelines. However, the study still needs to identify the precise primary incidents.

The work presented here differs from previous work in four important ways.

(1) Each crash is linked with an incident timeline, which is a necessary precursor to investigating the relationship between reliability and safety. Previous studies have conducted a statistical test with traffic volume and accident frequency.

(2) A novel methodology is proposed to classify crashes in recurrent congestion as well as crashes in non-recurrent congestion.

(3) The "recurrent" bottleneck identification approach is used to identify recurrent bottleneck locations and their impact areas.

(4) The methodology does not require any information on primary incidents (crashes and non-crash incidents) to identify crashes in non-recurrent congestion.

## 3. Methodology

A four-step methodology is employed, as depicted in Figure 1. In Step 1, both mobility and crash data are directionally and temporally linked to freeway segments using a GIS-based shapefile. Fifteen minute speed data are used as a mobility surrogate. In Step 2, the crashes are classified based on whether they are occurring during normally congested or uncongested periods. To differentiate

between these conditions, a cut-off threshold is employed. In Step 3, for the crashes that occur during normally uncongested periods, it is determined whether non-recurring congestion was present at the time of the crash. Crashes occurring in non-recurring congestion where no or little bottleneck is normally activated are classified with a specific cut-off threshold. This threshold comes from a historic spatiotemporal congestion index that identifies recurrent congestion. Finally, in Step 4, the remaining crashes are classified as being during recurrent or non-recurrent conditions. First, crashes are classified as being recurrent congestion if they are within a recurrent congestion impact area whose bottleneck activates frequently. Crashes that cannot be classified this way are treated as special cases and classified with a supplemental methodology.

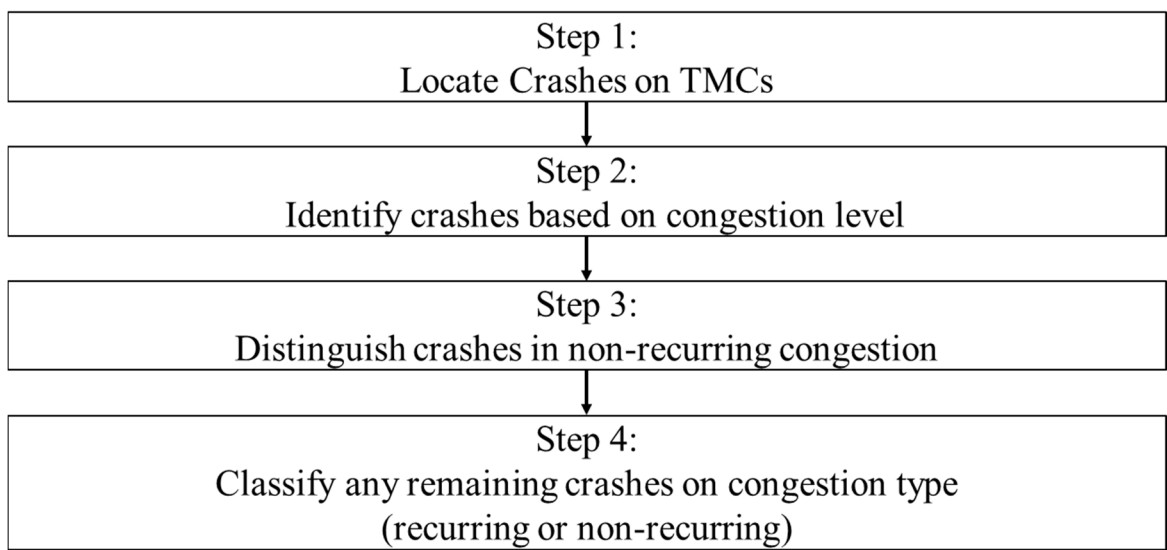

**Figure 1.** Methodology flow procedure.

### 3.1. Data Sources

#### 3.1.1. Mobility Data

Speed data from INRIX.com was employed in the study. INRIX.com uses GPS enabled probe vehicles to collect this information. The geocoding is based on Traffic Message Channel (TMC) codes, as defined by Tele Atals and Navteq. Each TMC corresponds to a directional roadway segment with geolocated begin and end points. INRIX reports average travel times, average speeds, reference speeds, and scores by time of day and day of the week. The score indicates if the reported speed is based on historical data, real time data, or a blend of the two.

In the study area, the Regional Integrated Transportation Information System (RITIS) provides an integrated spatiotemporal contour map of the traffic congestion for each TMC segment by time of day (RITIS) [33]. RITIS also provides contour maps for comparative speeds, congestion, a travel time index, and so forth. This study used 15 min. aggregated congestion data. RITIS defines the congestion value, $C(i,t,m)$, for segment ($i$) at time ($t$). $i$ is TMC a segment and $t$ is specified time interval in a day (e.g., 8:00–8:15, 15 min.) and typically will vary from 1 to 96 for a single day. $m$ is index for a day in the study period especially for weekday. Finally, $MS(i, t, m)$, the measured speed (mi/h), and $FFS(i)$ the free flow speed (mi/h) are specified for each segment.

Figure 2a shows sample congestion data obtained directly from RITIS.org. The TMC segment is shown on the vertical axis and time on the horizontal axis. Congestion values are shown in the cells. Traffic flow direction is from bottom to top from TMC segment 125-10218 to TMC segment 125N04645. Figure 2b presents sample congestion index (CI) data.

Congestion for I-85 between Statesville Ave/Exit 39 and Kannap
Southbound June 11, 2014

| TMC CODE | NAME | MILES | 3:00 PM | 3:15 PM | 3:30 PM | 3:45 PM | 4:00 PM | 4:15 PM | 4:30 PM | 4:45 PM | 5:00 PM | 5:15 PM | 5:30 PM |
|---|---|---|---|---|---|---|---|---|---|---|---|---|---|
| 125N04645 | W Mallard Creek Church Rd/Exit 46 | 0.60 | 100% | 100% | 100% | 100% | 100% | 100% | 100% | 100% | 94% | 92% | 91% |
| 125-04645 | W Mallard Creek Church Rd/Exit 46 | 1.08 | 100% | 100% | 94% | 96% | 93% | 94% | 97% | 93% | 92% | 98% | 92% |
| 125N04646 | I-485/Exit 48 | 0.35 | 100% | 79% | 40% | 42% | 58% | 41% | 52% | 41% | 42% | 38% | 52% |
| 125-04646 | I-485/Exit 48 | 0.94 | 100% | 75% | 17% | 11% | 17% | 15% | 21% | 15% | 35% | 32% | 41% |
| 125N04647 | Speedway Blvd/Exit 49 | 0.73 | 100% | 100% | 78% | 18% | 9% | 12% | 15% | 15% | 34% | 40% | 57% |
| 125-04647 | Speedway Blvd/Exit 49 | 1.82 | 100% | 100% | 100% | 71% | 32% | 15% | 10% | 8% | 20% | 49% | 99% |
| 125N04648 | Poplar Tent Rd/Exit 52 | 0.53 | 100% | 100% | 100% | 99% | 100% | 99% | 72% | 13% | 13% | 59% | 100% |
| 125-04648 | Poplar Tent Rd/Exit 52 | 1.32 | 100% | 100% | 100% | 100% | 100% | 99% | 100% | 83% | 35% | 74% | 100% |
| 125N10218 | Kannapolis Pkwy/Exit 54 | 0.70 | 100% | 100% | 100% | 100% | 100% | 100% | 100% | 100% | 100% | 100% | 100% |
| 125-10218 | Kannapolis Pkwy/Exit 54 | 0.83 | 100% | 100% | 100% | 100% | 100% | 100% | 100% | 100% | 100% | 100% | 100% |

(**a**) Congestion value

Congestion for I-85 between Statesville Ave/Exit 39 and K
Southbound June 11, 2014

| TMC CODE | NAME | MILES | 3:00 PM | 3:15 PM | 3:30 PM | 3:45 PM | 4:00 PM | 4:15 PM | 4:30 PM | 4:45 PM | 5:00 PM | 5:15 PM | 5:30 PM |
|---|---|---|---|---|---|---|---|---|---|---|---|---|---|
| 125N04645 | W Mallard Creek Church Rd/Exit 46 | 0.60 | 0 | 0 | 0 | 0 | 0 | 0 | 0 | 0 | 0 | 0 | 0 |
| 125-04645 | W Mallard Creek Church Rd/Exit 46 | 1.08 | 0 | 0 | 0 | 0 | 0 | 0 | 0 | 0 | 0 | 0 | 0 |
| 125N04646 | I-485/Exit 48 | 0.35 | 0 | 1 | 1 | 1 | 1 | 1 | 1 | 1 | 1 | 1 | 1 |
| 125-04646 | I-485/Exit 48 | 0.94 | 0 | 1 | 1 | 1 | 1 | 1 | 1 | 1 | 1 | 1 | 1 |
| 125N04647 | Speedway Blvd/Exit 49 | 0.73 | 0 | 0 | 1 | 1 | 1 | 1 | 1 | 1 | 1 | 1 | 1 |
| 125-04647 | Speedway Blvd/Exit 49 | 1.82 | 0 | 0 | 0 | 1 | 1 | 1 | 1 | 1 | 1 | 1 | 0 |
| 125N04648 | Poplar Tent Rd/Exit 52 | 0.53 | 0 | 0 | 0 | 0 | 0 | 0 | 1 | 1 | 1 | 1 | 0 |
| 125-04648 | Poplar Tent Rd/Exit 52 | 1.32 | 0 | 0 | 0 | 0 | 0 | 0 | 0 | 0 | 1 | 1 | 0 |
| 125N10218 | Kannapolis Pkwy/Exit 54 | 0.70 | 0 | 0 | 0 | 0 | 0 | 0 | 0 | 0 | 0 | 0 | 0 |
| 125-10218 | Kannapolis Pkwy/Exit 54 | 0.83 | 0 | 0 | 0 | 0 | 0 | 0 | 0 | 0 | 0 | 0 | 0 |

(**b**) Congestion Index

**Figure 2.** Traffic flow contours: (**a**) congestion value; (**b**) congestion index contour.

### 3.1.2. Crash Data

The North Carolina Department of Transportation maintains a database called the Traffic Engineering Accident Analysis System (TEAAS) for traffic crashes. Records in the TEAAS database indicate injury severity, roadway characteristics, crash characteristics and location, and environmental characteristics in terms of a set of prespecified categories (NCDOT) [3]. This study timeframe included a total of 1632 crashes for Tuesdays, Wednesdays, and Thursdays in April, May, September, and October of 2012 and 2013 across the 274 km study section of I-40 in North Carolina. Table 1 presents two examples of the TEAAS crash data.

**Table 1.** The sample of Traffic Engineering Accident Analysis System (TEAAS) crash data.

| Crash ID | Vehicle Type | Severity | Direction | Crash Day | ... | Crash Type | Locality | Weather |
|---|---|---|---|---|---|---|---|---|
| 1034169 | VEH | O (No injury) | Westbound | Tuesday | ... | Rear-end | Rural | Clear |
| 4891526 | CMV | K (death) | Northbound | Thursday | ... | Animal | Urban | Wet |
| : | : | : | : | : | : | : | : | : |

### 3.2. Congestion Assessment Methodology

A method was developed to determine the level of congestion that was extant at the time of each crash. As mentioned earlier, the recurrent bottleneck identification approach developed by Song et al. [11] was employed. It makes use of a Congestion Index (CI), an Average Historic Congestion Index (AHCI), and a Recurrent Bottleneck Location Identification (RBLI).

The congestion index, $CI(i, t, m)$, labels each segment $i$ as being congested or uncongested at time $t$, as illustrated in Figure 2b. The congestion value for each segment is determined. If the congestion value is below a threshold $\alpha$, segment ($i$) is classified as being congested at time $t$. Here, a value of 80% was used as the threshold. This value is consistent with the ratio of (the speed at capacity) to (the speed at free flow) presented in the US Highway Capacity Manual [34].

The Average Historic Congestion Index, $AHCI(i, t)$, is defined as the faction of (week or all) days in the study period $T$ (typically one or two years) where segment $i$ was congested at time $t$, based

on the specified congestion index (($CI(i, t, m)$)). *AHCI* is the key parameter for identifying recurrent congestion. It is used to denote the probability that segment $i$ was congested at time $t$.

AHCI contour maps are used to define the recurrent bottlenecks as well as their influence areas. To illustrate, Figure 3 shows three different patterns that were observed in Spring and Fall seasons of 2013 in North Carolina. Each block corresponds to an AHCI value at a TMC segment at a certain time of day. A TMC with the highest AHCI (more than 50%) value a given time of day is defined as a recurrent bottleneck segment, $\beta$, which was proposed from Song et al. [11].

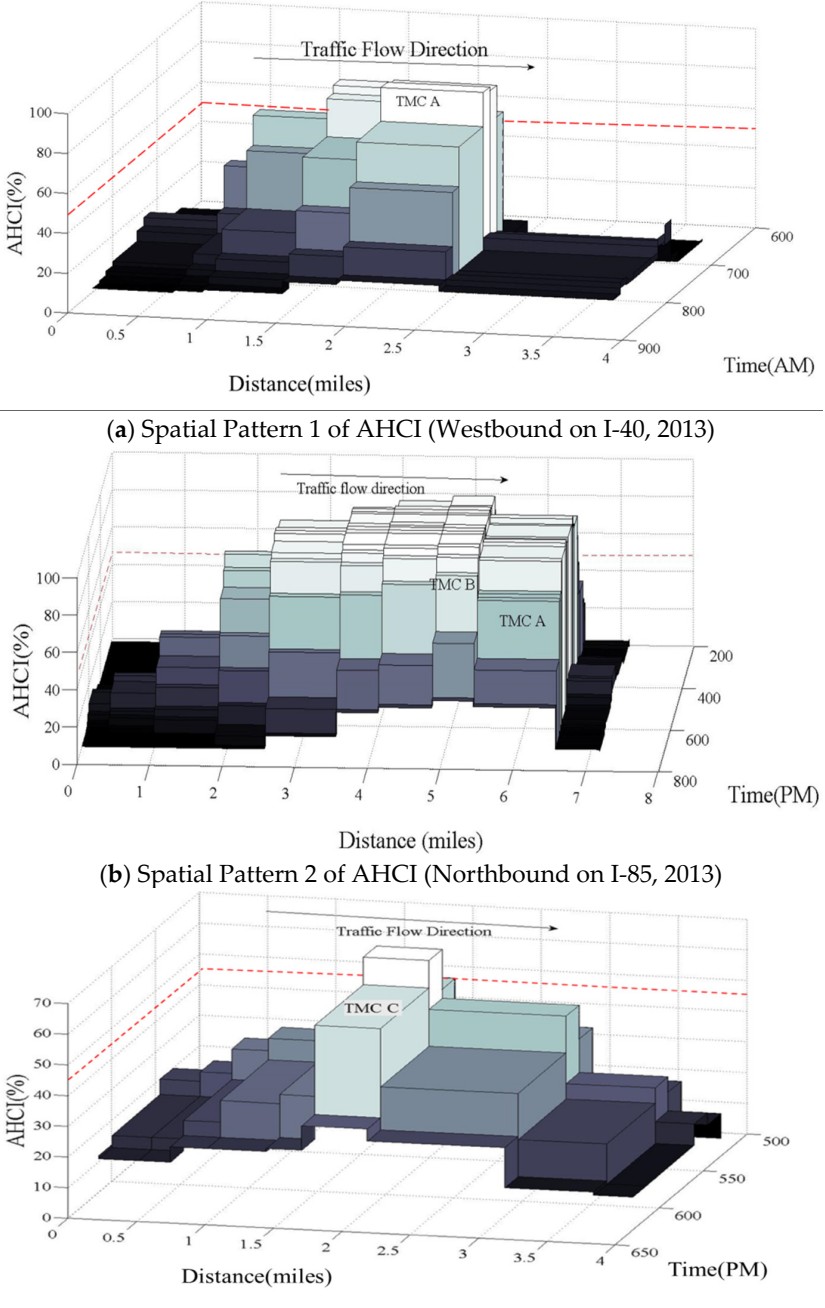

(**a**) Spatial Pattern 1 of AHCI (Westbound on I-40, 2013)

(**b**) Spatial Pattern 2 of AHCI (Northbound on I-85, 2013)

(**c**) Spatial Pattern 3 of AHCI(Eastbound on I-40, 2013)

**Figure 3.** Spatial patterns of Average Historic Congestion Index (AHCI): (**a**) Spatial Pattern 1 of AHCI (Westbound on I-40, 2013); (**b**) Spatial Pattern 2 of AHCI (Northbound on I-85, 2013); (**c**) Spatial Pattern 3 of AHCI (Eastbound on I-40, 2013).

The question then is what other nearby TMC segments are also in a recurrent bottleneck area? In reference to Figure 3a, the TMC with the highest AHCI above 50% (TMC A) is located just upstream of a significant drop in AHCI value. Therefore, this TMC segment is clearly in a recurrent bottleneck and this pattern is defined as Pattern 1. In Figure 3b, the TMC segment with the highest AHCI value (TMC B) is not located immediately upstream of a significant drop in the AHCI value. However, TMC B is also part of the bottleneck area and this pattern is defined as Pattern 2. In Figure 3c, the TMC segment with the highest AHCI value (greater than 50%) is in the bottleneck area (TMC C); however, a significant drop is not observed in the AHCI value for this TMC segment or the one downstream of it. In addition, the downstream TMC segment has an AHCI value below 50%. Therefore, the bottleneck area is defined by only TMC C and this pattern is defined as Pattern 3. In this study, the bottleneck influence area includes all the TMC segments upstream of a bottleneck area with AHCI values greater than 20%, with the assumption that congestion occurs on at least one of every five weekdays. The cut-off threshold of boundary of bottleneck influence area, $\gamma$, is reasonable in that spatial patterns depicts all TMC segments is located upstream have greater AHCI value than downstream TMCs' as seen in Figure 3a–c.

The recurrent bottleneck identification methodology developed by Song et al. [11] employs an exhaustive search algorithm to identify the recurrent bottleneck locations. Two constraints are imposed on this search: First, to be included in a recurrent bottleneck location, all contiguous TMC segments must have AHCI values exceeding 50%. Second, at most two of the TMC segments must be included by spatial pattern 2. A threshold, $\delta$, is defined as the allowable difference in AHCI. The proposed $\delta$ value in this study is 2. In Figure 4, the TMC U segment produces an AHCI value greater than $\delta$, which is greater than the AHCI value for TMC Ds, so this segment is included in the recurrent bottleneck. Song et al. [11] recommended values of $\delta$ equal to 2.0, 2.4, and 2.5, which can be selected differently by transportation jurisdictions. The methodology is as follows Algorithm 1. The recurrent bottleneck identification methodology.

---

**Algorithm 1.** The recurrent bottleneck identification methodology.

---

*For all each spatiotemporal AHCI(i, t)*
　　*If AHCI(i, t) ≥ 50%*
　　*Spatial pattern 1, 3: $y_1 = AHCI(i, t) - 2 \cdot AHCI(i+1, t)$*
　　*Spatial pattern 2(1): $y_2 = AHCI(i, t) - 2 \cdot AHCI(i+2, t)$*
　　*Spatial pattern 2(2): $y_3 = AHCI(i, t) - AHCI(i+1, t)$*
　　*If $y_3 \geq 0$ && $y_1 \geq 0 || y_2 \geq 0$*
*Then, the segment (i) possibly as bottleneck*

---

| TMC Segment | Name | Miles | 7:15AM | 7:30AM | 7:45AM | 8:00AM | 8:15AM | 8:30AM | 8:45AM | 9:AM |
|---|---|---|---|---|---|---|---|---|---|---|
| 125P04965 | Gorman St/Exit 295 (TMC D) | 0.38 | 1% | 4% | 15% | 17% | 11% | 11% | 12% | 9% |
| 125+04965 | Gorman St/Exit 295 (TMC U) | 1.26 | 7% | 72% | 88% | 90% | 73% | 53% | 26% | |
| 125P04964 | Lake Wheeler Rd/Exit 297 | 0.74 | 8% | 67% | 87% | 83% | 64% | 40% | 21% | |
| 125+04964 | Lake Wheeler Rd/Exit 297 | 0.39 | 6% | 40% | 70% | 55% | 26% | 14% | 8% | |
| 125P04963 | US-70/US-401/Exit 298 | 0.58 | 5% | 16% | 44% | 23% | 13% | 8% | 5% | |
| 125+04963 | US-70/US-401/Exit 298 | 0.2 | 5% | 6% | 16% | 12% | 7% | 5% | 3% | 2% |

**Figure 4.** Bottleneck segment identification using an AHCI contour map.

### 3.3. Collision Classification Methodology by Type of Congestion

A four-step method is used to add a congestion level indicator to the crash records, as shown in Figure 5. This section provides detailed explanations for each step.

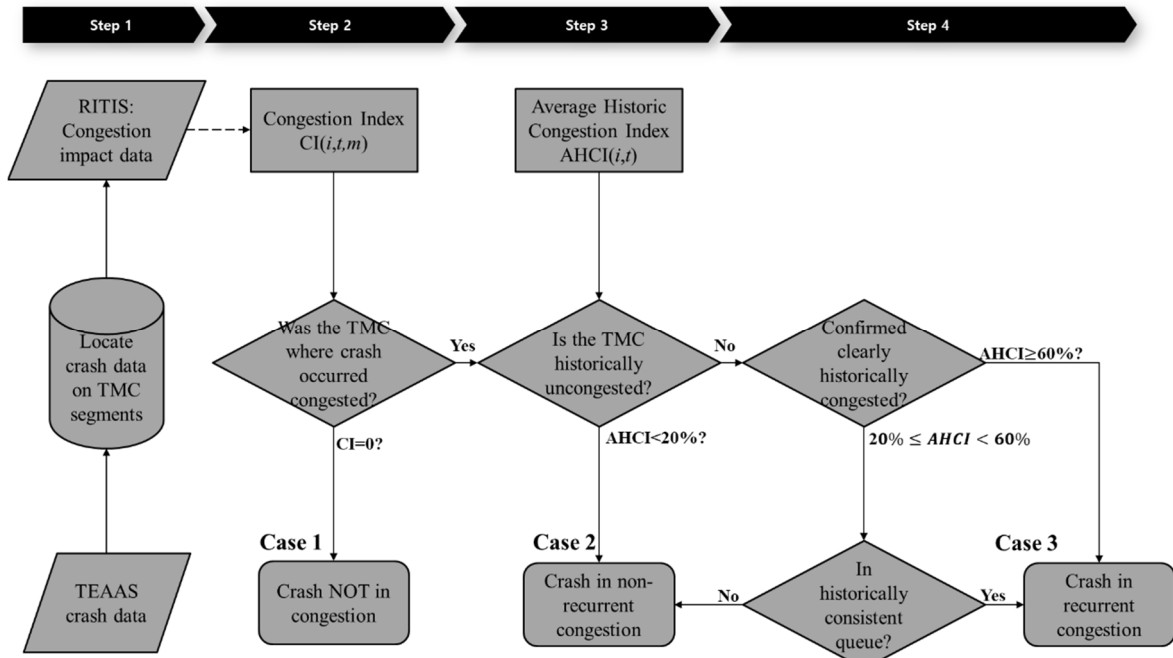

**Figure 5.** The collision classification procedure.

### 3.3.1. Step1: Locate the Crash in GIS

First, a TMC code is associated with the crash. As the crash records include longitude and latitude information, the TMC codes can be identified easily. A link shapefile of TMC segments is applied to join the two datasets. Thus, crash becomes associated with specific TMC segment, which is defined as "TMCc."

### 3.3.2. Step2: Crashes in Uncongested Conditions

The second step is to identify the crashes which occur during uncongested conditions. As shown in Figure 5, if the crash occurs on a TMC segment that is uncongested at the time of the crash, then the crash is labeled as "Case 1—Crash not in congested conditions". Otherwise, the analysis continues to Step 3. As mentioned earlier, if $CI(i,t,m) = 0$, then TMC segment $i$ is considered to be uncongested at time $t$ and vice versa.

### 3.3.3. Step3: Crashes in Congested Conditions

For those crashes that do not occur during uncongested conditions, the third step is to determine the congestion status of the TMC segment at the time of the crash. Two main metrics are employed: $CI(i,t,m)$ and $AHCI(i,t)$. The analysis focuses on the pattern $AHCI(i,t)$ values, during the crash versus "normally". As mentioned earlier, two breakpoints, $\beta$ and $r$, are used to evaluate $AHCI(i,t)$. If $AHCI(t) < \gamma$, then TMC segment $i$ is considered to be historically uncongested at time $t$; otherwise, if $AHCI(t) \geq \beta$, then it is historically congested.

The values of $\beta$ and $r$ are determined through a sensitivity analysis. Figure 6 shows the sensitivity the sensitivity analysis for crashes labeled as "Case 2—Crash in non-recurrent congestion". It is clear that if the upper bound is between 50 and 70 and the lower bound is between 10 and 20, the number of Case 2 selections becomes quite stable. However, if the upper bound drops below 50 or the lower

bound is increased beyond 30, the number of Case 2 classifications changes significantly. This leads to the conclusion that the upper and lower bounds used in the selection process presented above are a good combination to employ. A value of 60% as the threshold of $\beta$ and 20% for $\gamma$. The values of $\beta$ and $r$ were 20% and 60%, respectively. Therefore, if *AHCI*(*t*), then TMC segment *i* is congested at time *t* less than once in every five days. If *AHCI*(*t*)$\geq$ 60%, then it is congested at least every other day. Therefore, if *AHCI*(*t*) < $\gamma$, then the crash is labeled as "Case 2". Otherwise, it is passed to Step 4.

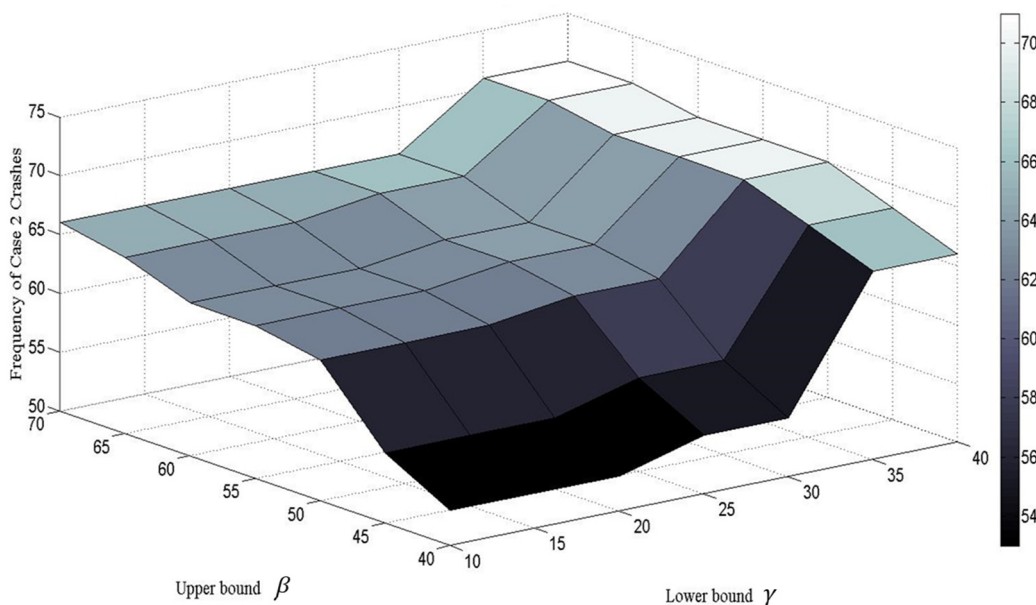

**Figure 6.** Number of Case 2 under various upper and lower bound of AHCI.

### 3.3.4. Step 4: Classifying Remaining Crashes (Supplemental Methodology)

In step four, if *AHCI*(*t*) $\geq$ $\beta$, then the crash is labeled "Case 3—Crash in recurrent congestion". Otherwise (that is, 20% < AHCI $\leq$ 60%), then a sub-test is performed to see if recurrent congestion is typically present at the time of the crash. If recurring congestion is typically present, then the crash is labeled "Case 3". Otherwise, it is labeled "Case 2". In the area of uncertainty given by 20% < AHCI $\leq$ 60%, a supplemental test is made to see whether there is a recurrent bottleneck TMC downstream of crash location, and whether the queue from that TMC typically spills back into the crash location. This algorithm is seen in Algorithm 2.

---

**Algorithm 2.** Classifying Remaining Crashes (Supplemental Methodology).

---

*If TMCc is in a recurrent bottleneck segment*
　　*Go to Case 3*
*IElseif TMCc is not in a recurrent bottleneck segment*
　　　$AHCI_{TMC_C}(i,t) \leq AHCI_{TMC_C}(i+1,t)$*&&*
$AHCI_{TMC_C}(i,t) \leq AHCI_{TMC_C}(i+2,t)$*&&*
　　　　$\vdots$
$AHCI_{TMC_C}(i,t) \leq AHCI_{TMC_C}(i+2,t)$
　　　*Go to Case 2*
　　*Else Go to Case 2*

---

In the algorithm, *AHCI*$_{TMCc}$(*i*,*t*) is the AHCI of TMCc, and *AHCI*$_{TMCc}$(*i+b*, *t*) is the AHCI of the recurrent bottleneck location downstream of TMCc.

## 4. Illustrated Case Study

This section illustrates the use of the classification methodology. Data from the 274 km "test site" is employed.

### 4.1. Study Site

The 274 km section of the I-40 used to test the methodology extends from Exit 259 (at the split with I-85 in Durham) to Exit 420 (Gordon Rd, at the eastern end of I-40 outside of Wilmington). This section contains 98 TMC segments, with an average length of 1.6 miles and a standard deviation of 2.05 miles. The first, median, and the third quartile values are 0.51, 0.67, and 1.56, respectively. The posted speed limit is either 65 or 70 mph. The data employed were for Tuesdays, Wednesdays, and Thursdays in April, May, September, and October of 2012 and 2013 (a total of 105 days). Aggregated 15-min TMC segment data were used to create the congestion contour plots. The crash data were for both directions. The crash dataset comprised 500 records (234 crashes westbound and 266 eastbound).

### 4.2. Crash Classification Results

Crashes on the TMC segments were classified using the methodology described Section 3. Figure 7 shows the contour map for one of the crashes classified "Case 1—Crash not in congested conditions". The number of Case 1 crashes was 419 out of 500 (i.e., 84%).

| TMC Segment | Rd. Name | Miles | 7:30PM | 7:45PM | 8:00PM | 8:15PM | 8:30PM | 8:45PM | 9:00PM | 9:15PM | 9:30PM |
|---|---|---|---|---|---|---|---|---|---|---|---|
| 125N04860 | Exit 287 | 0.6 | 0 | 0 | 0 | 0 | 0 | 0 | 0 | 0 | 0 |
| 125-04860 | Exit 287 | 1.7 | 0 | 0 | 0 | 0 | 0 | 0 | 0 | 0 | 0 |
| 125N04861 | Exit 285 | 0.7 | 0 | 0 | 0 | 0 | 0 | 0 | 0 | 0 | 0 |
| 125-04861 | Exit 285 | 0.5 | 0 | 1 | 1 | 1 | 1 | 1 | 0 | 0 | 0 |
| 125N04862 | Exit 284 | 0.7 | 0 | 1 | 1 | 1 | 1 | 1 | 1 | 0 | 0 |
| 125-04862 | Exit 284 | 0.6 | 0 | 0 | 1 | 1 | 1 | 1 | 1 | 0 | 0 |
| 125N04863 | Exit 283 | 0.7 | 0 | 0 | 1 | 1 | 1 | 1 | 0 | 0 | 0 |
| 125-04863 | Exit 283 | 0.3 | 0 | 0 | 1 | 1 | 1 | 0 | 0 | 0 | 0 |
| 125N04864 | Exit 282 | 0.2 | 0 | 0 | 0 | 0 | 0 | 0 | 0 | 0 | 0 |

**Figure 7.** Schematic representative a Case 1 crash condition.

In Step 3, the remaining 81 crashes were further analyzed. Of them, 62 were classified as belonging to Case 2 (a Case 2 is one that occurs in non-recurrent congestion). Panels (a) and (b) of Figure 8 show a crash that was identified as belonging to Case 2.

Figure 8e shows an instance where the Case 3 classification was based on the CI contour map. Figure 8f presents an instance where the Case 3 classification was based on the AHCI contour map.

The remaining nine crashes were passed to Step 4. Of these, nine had an AHCI value of more than 60% and were placed in Case 3. Panels (c) and (d) of Figure 8 show an example of a TMCc($AHCI(t) \geq 60\%$) that was identified as belonging to Case 3. The remaining 10 crashes were identified as being moved on to the sub-test in Step 4 and were subsequently also classified as Case 3.

The results of the crash classifications are summarized in Table 2. As can be seen, the proportions of cases 1, 2, and 3 were 84%, 12%, and 4%.

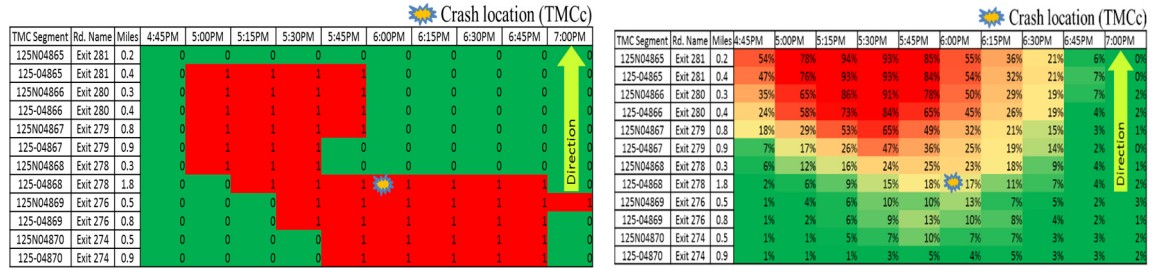

(**a**) A TMCc classified as Case 2 in CI (congestion)

(**b**) A TMCc classified as Case 2 in AHCI (non-recurrent)

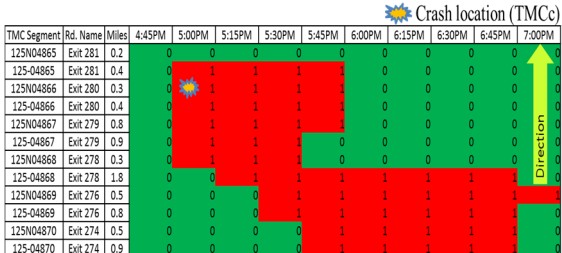

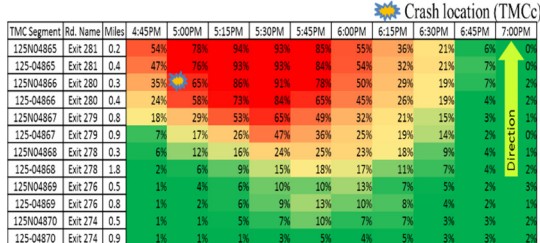

(**c**) A TMCc classified as Case 3 in CI (congestion)

(**d**) A TMCc classified as Case 3 in AHCI (recurrent)

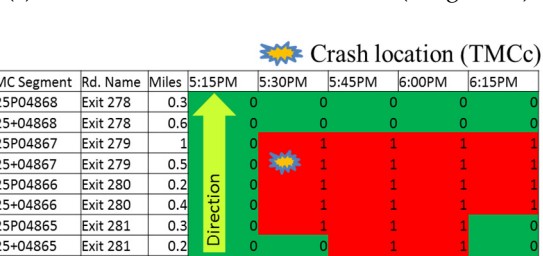

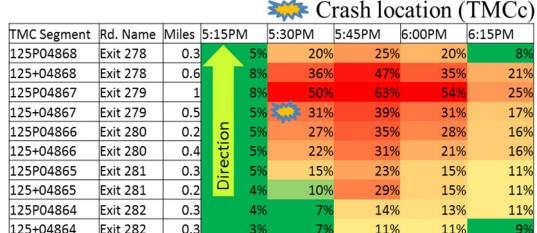

(**e**) A TMCc classified as Case 3 in CI (congestion) by the sub-test

(**f**) A TMCc classified as Case 3 in AHCI (recurrent) by the sub-test

**Figure 8.** Examples of TMCc classified as Cases 2 and 3 in Steps 3 and 4: (**a**) A TMCc classified as Case 2 in CI (congestion); (**b**) A TMCc classified as Case 2 in AHCI (non-recurrent); (**c**) A TMCc classified as Case 3 in CI (congestion); (**d**) A TMCc classified as Case 3 in AHCI (recurrent); (**e**) A TMCc classified as Case 3 in CI (congestion) by the sub-test; (**f**) A TMCc classified as Case 3 in AHCI (recurrent) by the sub-test.

**Table 2.** Classification of the crashes based on the proposed methodology.

| | I-40 | Westbound | Eastbound | Total |
|---|---|---|---|---|
| | Number of crashes | 234 | 266 | 500 |
| Step 2 | Case 1 (crash not in congestion) | 192 (82%) | 227 (85%) | 419 (84%) |
| Step 3 | Case 2 (crash in non-recurrent congestion) | 36 (15%) | 26 (10%) | 62 (12%) |
| Step4 | Case 3 (crash in recurrent congestion) | 6 (1%) | 13 (3%) | 19 (2%) |
| | To algorithm for congestion type classification | 3 (1%) | 7 (3%) | 10 (2%) |
| | Case 2 (crash in non-recurrent congestion) by the sub-test | 0 (0%) | 0 (0%) | 0 (0%) |
| | Case 3 (crash in recurrent congestion) by the sub-test | 3 (1%) | 7 (3%) | 10 (2%) |

## 5. Comparative Analysis

Table 3 shows the proportion of reported incidents or crashes and those not reported for the 62 samples of crashes in non-recurrent congested TMC's. As seen in Table 3, in only 58% of the cases was a primary crash or incident also reported. This indicates that more crashes in a non-recurrent congested area will be detected by the use of the proposed methodology. Otherwise, only those crashes tied to a primary incident will be classified, in our case 39 of the possible 62.

**Table 3.** Reported vs. unreported primary incidents or crashes on crash in non-recurrent congested areas.

| Crashes in Non-Recurrent Congestion Locations | Reported Incident or Crash | | No Data Primary Incident or Crash | Total Crashes in Non-Recurrent Congested TMC |
|---|---|---|---|---|
| | Primary Crash | Primary Incident | | |
| Number of crashes | 36 | 3 | 23 | 62 |
| Percentage | 58% | 5% | 37% | 100% |

## 6. Conclusions

This paper has presented a method for classifying crashes based on the type of congestion in which they occur. The methodology has been tested using North Carolina crash data for 274 centerline kilometers on I-40 in North Carolina for Tuesdays, Wednesdays, and Thursdays in April, May, September, and October of 2012 and 2013. In addition, an approach for identifying "recurrent" congestion has been used as part of the procedure. Unlike previous studies that used the mean and median of the speed distribution to distinguish recurrent and non-recurrent congestion pattern, the method used here employs a recurring congestion definition that is based on an average congestion history using probe-based speeds.

The proportion of secondary crashes (a surrogate for crashes in non-recurrent congestion) identified in the case study is in line with results from previous classification studies, where the secondary crash percentage ranged from 2.2% to 15.5% [28,35–41]. However, the study found a secondary crash proportion that is near the upper end of the range reported in these earlier studies. This is to be expected because the proposed methodology classifies crashes as secondary whenever they occur in atypical congestion without the need to identify the primary crash or incident event.

There are some limitations to the study. The most important one is the need to validate or verify efforts regarding secondary incidents or crash identification. It calls for identifying and defining real-world secondary events with more detailed approaches in the real world. Another issue is that the link-based traffic data provide uniform traffic performance information for the entire TMC. Thus, this study employed the starting and ending points of TMC's as the beginning and ending points for the congestion. This may cause fewer errors in identifying non-recurrent congestion conditions because of the limitation of the TMC segment itself (different link lengths). However, this limitation can be addressed as vendors provide speed data for sub-segments. In addition, as the probe percentages increase, data quality will also be an issue.

Despite the fact that only 4% of the total crashes in this case study were identified as a "Crash in recurrent congestion", the percentage of "secondary" crashes caused by primary incidents in recurrent congestion increased compared well to the results of previous studies. Therefore, identifying these crashes calls for a further detailed classification methodology, which the authors are presently investigating with more accurate crash data available from the state-of-the-art technologies of vehicle such as vehicle black box and event data recorder can be used. Finally, there has been no consideration about the impact of rubbernecking because the objective of this study was to focus on developing for a robust and easily implementable methodology to classify crashes in different types of congestion.

**Author Contributions:** Conceptualization, T.-J.S., S.K., B.M.W., and N.M.R.; methodology, T.-J.S.; formal analysis, T.-J.S. and B.M.W.; investigation, T.-J.S.; resources, B.M.W.; data curation, T.-J.S. and B.M.W.; writing—original draft preparation, T.-J.S. and S.K.; writing—review and editing, B.M.W., N.M.R., and G.F.L.; visualization, T.-J.S.; supervision, B.M.W., N.M.R., and G.F.L.; project administration, T.-J.S.; funding acquisition, B.M.W. All authors have read and agreed to the published version of the manuscript.

**Funding:** This research was funded by the North Carolina Department of Transportation (NCDOT), grant number NCDOT Research Project 2014-12.

**Conflicts of Interest:** The authors declare no conflicts of interest.

## Glossary/Nomenclature/Abbreviations

| | |
|---|---|
| $\alpha$ | cut-off threshold to determine being congested |
| $\beta$ | cut-off threshold to determine recurrent bottleneck segment |
| $\gamma$ | cut-off threshold of boundary of bottleneck influence area |
| $\delta$ | cut-off threshold of spatial congestion difference in AHCI |
| $AHCI(i,t)$ | average historic congestion index |
| $C(i,t,m)$ | congestion value |
| $CI(i,t,m)$ | congestion index |
| $FFS(i)$ | free flow speed (mi/h) |
| $i$ | a TMC segment |
| $m$ | index for a day in the study period (weekday) |
| $MS(i,t,m)$ | measured speed (mi/h) |
| RITIS | Regional Integrated Transportation Information System (RITIS) |
| $t$ | time interval a day (e.g., 8:00–8:15, 15 min) |
| TEAAS | Traffic Engineering Accident Analysis System |
| TMC | Traffic Message Channel |
| TMCc | TMC segment associated with a crash |

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
