# Peer review of "Crash Classification by Congestion Type for Highways"

_applsci, doi:10.3390/app10072583_

Round 1

Reviewer 1 Report

  1. It is not quite explicit how the work advances state-of-the-art research. Although some discussion has been provided in the Related work section; it does not seem sufficient to reveal the novel contribution of this work.
  2. The novelty of the proposed methodology should be emphasized.
  3. Most of the references are old. Authors should refer to more recent publications to reflect the cutting-edge research results.
  4. When presenting a methodology or method, the possible alternatives should be discussed so that the design decisions can be justified and better understood.

Author Response

Dear reviewer, 

Please consider our revised manuscript for the publication in Applied Science, with a new title as "Crash Classification by Congestion Type for Highways".

We are very grateful to the reviewer for the valuable comments and thoughtful handling of the manuscript, which have undoubtedly helped to improve the substance and readability of our paper. All of the reviewer's comments and suggestions have been addressed, and the manuscript has been revised substantially both in terms of the technical content and English language. We appreciate the opportunity to make use of these constructive comments to strengthen the paper. In this memo, we have addressed each of the reviewer's comments on a point-by-point basis, and all of the revised parts have been highlighted in red.  

Reviewer 2 Report

Dear Authors:

Please find my comments in the attached file.

Best Regards

Author Response

(The authors gave the same response as above.)

Round 2

Reviewer 1 Report

The introduction can be enriched; currently, it is too short.

There can be a more detailed description of the overall methodology.

Author Response

Dear reviewers, 

Thanks to the reviewers valuable comments, our paper get improved the substance and readability. All of the reviewers' comments and suggestions have been addressed, and the manuscript has been revised substantially both in terms of the technical content and English language.

All the best, 

Corresponding author,

Sangkey Kim, Ph.D.
